MSBOTS: a multiple small biological organism tracking system robust against non-ideal detection and segmentation conditions

Wang Xiaoying 1 xiaoying.wang@rmit.edu.au
http://orcid.org/0000-0003-1632-8062 Cheng Eva 2
Burnett Ian S. 2
1 School of Engineering, RMIT University , Melbourne , Australia
2 Faculty of Engineering & Information Technology, University of Technology Sydney , Sydney, New South Wales , Australia
Scheibe Timothy
Electronic publication date: 2021 Jul 27
Publication date: 2021
Volume: 9
Electronic Location ID: e11750
Received 2020 Nov 25; Accepted 2021 Jun 19
Copyright: © 2021 Wang et al.
Copyright year: 2021
Copyright holder: Wang et al.
License: This is an open access article distributed under the terms of the Creative Commons Attribution License, which permits unrestricted use, distribution, reproduction and adaptation in any medium and for any purpose provided that it is properly attributed. For attribution, the original author(s), title, publication source (PeerJ) and either DOI or URL of the article must be cited.
License URL: https://creativecommons.org/licenses/by/4.0/

Keywords: Automatic multiple organism tracking, Segmentation, Complex imaging conditions, Kuhn–Munkres algorithm

Funding: The authors received no funding for this work.

==============================
Accurately tracking a group of small biological organisms using algorithms to obtain their movement trajectories is essential to biomedical and pharmaceutical research. However, object mis-detection, segmentation errors and overlapped individual trajectories are particularly common issues that restrict the development of automatic multiple small organism tracking research. Extending on previous work, this paper presents an accurate and generalised Multiple Small Biological Organism Tracking System (MSBOTS), whose general feasibility is tested on three types of organisms. Evaluated on zebrafish, Artemia and Daphnia video datasets with a wide variety of imaging conditions, the proposed system exhibited decreased overall Multiple Object Tracking Precision (MOTP) errors of up to 77.59%. Moreover, MSBOTS obtained more reliable tracking trajectories with a decreased standard deviation of up to 47.68 pixels compared with the state-of-the-art idTracker system. This paper also presents a behaviour analysis module to study the locomotive characteristics of individual organisms from the obtained tracking trajectories. The developed MSBOTS with the locomotive analysis module and the tested video datasets are made freely available online for public research use.

Introduction

In recent years, small biological organisms such as zebrafish larvae (genetically and physiologically similar to humans), Artemia franciscana, and Daphnia magna have become powerful models and are widely used to study human disease (James et al., 2019), pharmacology (Comeche et al., 2017) and ecotoxicology (James et al., 2019; Comeche et al., 2017; Poynton et al., 2007). Accurate tracking techniques are vital for understanding the biology and ecology underlying their movement (Martineau & Mourrain, 2013; Nema et al., 2016; Colwill & Creton, 2011; Alyuruk, Demir & Cavas, 2013; Ekvall et al., 2013). The traditional method relying on human visual observations is very tedious and time consuming. Also, the related experiments are difficult to reliably repeat. Though fluorescent labelling can improve the visual distinction of specific targets, fluorescent materials affect the behavioural response of these organisms (Ekvall et al., 2013).

Automatic object tracking techniques have assisted in developing approaches to the behaviour analysis of large organisms such as mammals, birds and adult fish. However, the tracking of small organisms are hampered by the constraints of existing automatic tracking systems (Ekvall et al., 2013; Dur et al., 2011) as most organisms are considerably smaller than 1 mm (Marechal et al., 2004). There are many challenges imposed by the small sized organisms. Firstly, the widely used transponder (also called u-chips) in individual organism identification dramatically affect the natural behaviour of organisms within millimetre-scales (Ekvall et al., 2013; Lard et al., 2010) when affixing to their bodies, because the currently available smallest chip size is around 0.4 mm (Usami, 2004; Rashid et al., 2012). In addition, general object tracking is already a complex problem due to object occlusion, non-rigid structure (caused by object rotation and scale changes), and motion pattern changes (Habibi, Sulistyaningrum & Setiyono, 2017). The challenge increases when the tracking targets are small in size, because small organisms provide very little information compared with imaging noise.

Existing automatic tracking systems for multiple organisms (Zhou et al., 2014; Conklin et al., 2015) either use adult subjects (Pérez-Escudero et al., 2014) in a large container to limit object interaction, or use a petri dish plate to separate individual objects, allowing only one object in each dish well to avoid overlapped and swapped trajectories. The machine learning based CNNTracker (Zhiping & Cheng, 2017) attempts to optimise individual identification accuracy by creating zebrafish head feature maps. However, this system only tested on adult zebrafish, which have very different movement characteristics and higher target-background imaging contrast than small sized larval organisms. Thus, this method may mis-classify larval organisms with multiple identities (Zhiping & Cheng, 2017). The Generalized Linear Mixed Model (GLMM) (Liu et al., 2017) studies the analysis of larval locomotive activities, but can only detect whether movement exists by reporting binary move/no-move classes, but does not enable object displacement estimation.

Accurate object detection and segmentation provides a critical foundation for the performance of the subsequent tracking process. However, the results of existing Multiple Target Tracking (MTT) algorithms degenerate caused by false positive segmentation (i.e., noise fragments remaining after segmentation) and false negative segmentation (i.e. mis-detected objects and occluded objects) (Mallick et al., 2013). Such segmentation challenges commonly occur in microscopic small organism videos when taken under realistic experimental conditions (without deliberate imaging control). It is nearly impossible to identify and segment all small organisms without picking up noise using these microscopic videos (Noss, Lorke & Niehaus, 2013). Non-ideal tracking results further pose challenges for behavioural analysis and maintaining individual identities over time (Martineau & Mourrain, 2013).

IdTracker (Pérez-Escudero et al., 2014) is a well-known biological organism tracking system with ‘fingerprint’ generation for subject identity differentiation, and the commercial LoliTrack (Závorka et al., 2017) system can also track multiple targets in a single container. However, both systems require the input videos to be taken under strict imaging conditions. As reported by Zhou et al. (2014) and Noss, Lorke & Niehaus (2013), even small impurities inside water (e.g., water bubbles) or lighting reflections (e.g., surface ripples) affect the object segmentation accuracy.

3D systems with multiple cameras or super-resolution images built from multiple low-resolution images are presented to obtain more information for accurate tracking of small organisms (Ekvall et al., 2013; Günel et al., 2019; Noss, Lorke & Niehaus, 2013). However, these systems increase computational complexity, change data association structure, and require further location registration among cameras or images. Thus, these challenges constrain the real-time application of these techniques in tracking multiple small-scaled organisms (Günel et al., 2019).

Extending on our previous work (Wang et al., 2017a, 2018), this paper presents an automatic and accurate Multiple Small Biological Organism Tracking System (MSBOTS). The approach is robust against non-ideal object detection and segmentation results that are obtained from microscopic time-lapse videos taken under practical laboratory experimental conditions. The system applies Gaussian Mixture Model (GMM) based background subtraction in the segmentation module (Wang et al., 2017a) to detect and segment small organisms from each video frame, and initially maps detected objects between successive video frames based on the (non-ideal) segmentation results using the Hungarian algorithm (also called Kuhn–Munkres algorithm) (Bourgeois & Lassalle, 1971; Munkres, 1957). The positions of mis-detected and overlapped objects are then calculated through their neighbour’s locations. Then the theoretically computed locations are bridged in the individual tracking trajectories (Wang et al., 2018). This paper extends on this existing work to present a novel behaviour analysis module to implement locomotive behaviour analysis such as estimating object velocities, accelerations and movement directions. The performance and versatility of MSBOTS is then evaluated and demonstrated by tracking accuracy, and compared with existing multiple organism tracking systems using a zebrafish larvae video dataset (Wang et al., 2017a) as used in our previous work. In this paper, to evaluate the generalisation of the proposed system to track other small-sized organisms, we also apply the system to video datasets of another two types of small organisms Artemia franciscana, and Daphnia magna.

Methods

Figure 1 outlines the overall workflow of the proposed MSBOTS platform, which comprises an object segmentation module (Wang et al., 2017a), data association module (Wang et al., 2018) and the behaviour analysis module novel to this paper. The accurate differentiation of organisms from the image background and foreign matter (such as water impurities and objects faeces) in each video frame is the critical foundation for this system. Here, the image background is estimated by an adaptive Gaussian Mixture Model (GMM) (Zivkovic & Heijden, 2006) in the segmentation module. Organisms in every video frame are segmented after the background subtraction; details of this segmentation approach can be found in our previous work (Wang et al., 2017a). The following data association module assigns detected objects (as sources, represented by the computed centroids of segmented regions) to their corresponding targets in the successive frames. This part plays an essential role in maintaining consistent individual identities for the detected organisms over time; details of this approach are in Wang et al. (2018). The data association (also called mapping) algorithm not only finds the most likely targets in the following frames for detected organisms, but also calculates the theoretical positions for mis-detected or occluded organisms. After obtaining the individual trajectories of all organisms from the output of the data association module, the movement characteristics of each organism are then estimated in the novel behaviour analysis module proposed in this paper as shown in Fig. 1.

Figure 1 Flow chart of the proposed MSBOTS platform.

It takes microscopic time-lapse frames as input, and consists of three main modules: object segmentation, data association based on the centroid locations of detected objects, and the behaviour analysis using the obtained individual trajectories.

This platform as a whole (as shown in Fig. 1) is tested on a zebrafish video dataset as used in our previous work (Wang et al., 2017a, 2018), and another two new video datasets of two different types of small organisms to evaluate the generalisation of our approach.

Code of ethics

Ethics approval is not required when filming videos of larval organisms. Further, no chemicals were tested with the larval organisms being filmed. The adult zebrafish video analysed is from a publicly available online repository (Pérez-Escudero et al., 2014).

Organism detection and segmentation

In the background subtraction step of the proposed MSBOTS platform, an improved GMM (Zivkovic & Heijden, 2006) was chosen to estimate the stationary background due to its ability to extend the detection period (Wang et al., 2017a). This is particularly effective for addressing the imbalanced movement problem of small biological organisms that occur in time-lapse microscopic video frames. As reported by Liu et al. (2017), small organisms such as zebrafish larvae exhibit a mean proportion of activities less than 7.5% over time with a ‘bursty’ movement pattern of sudden swimming motion interspersed with mainly stationary periods.

After video background estimation, the moving objects can be initially classified as the foreground organisms. Due to the ‘bursty’ movement pattern of small organisms, a moving organism will be firstly represented by a new region cluster with a small weight, whose value will be gradually increasing if the region remains stationary over time. This new region will not be classified as the background until its weight value exceeds a threshold cf, which is calculated as in Zivkovic & Heijden (2006). Thus, the detection period of a stationary organism can be extended for approximately log(1 − cf)(1 − α) frames, where α denotes the constant factor shaping an exponential decay envelope introduced in the GMM work by Zivkovic & Heijden (2006). This enables the detection and tracking of organisms that stop at a position for a certain period of time before restarting their motion.

Furthermore, to adapt to background changes in microscopic videos over time, the adaptive GMM (Zivkovic & Heijden, 2006) adds flexibility when selecting the number of Gaussian components and their parameters. The approach by Zivkovic & Van Der Heijden (2004) and the Dirichlet criteria are applied, respectively, in model initialisation and selection of the number of Gaussian components. The Gaussian component parameters are then adaptively updated for each video frame to accurately represent background pixels, in contrast to traditional GMM models that apply one or a fixed number of components. The foreground objects are then obtained by the differentiation of the video frames from the corresponding estimated background. Furthermore, post-processes such as median filtering, morphological grayscale erosion, and size-based noise removal are applied to eliminate image distortion and scattered noise fragments from the foreground. Within these, the morphological grayscale erosion deploys a flat diamond-shaped structuring element to erode the obtained organism foreground image. Thus, the boundaries among organisms in very close proximity are further widened; thus, the estimated centroid positions of the detected regions are more accurate.

The detection and segmentation of MSBOTS enables the removal of stationary backgrounds (such as the organism container and labels/markers drawn on its surface) and is robust against unbalanced organism movement patterns (e.g., ‘bursty’ movement) and water impurities. Hence, unlike existing systems, MSBOTS is able to process videos under practical experimental imaging conditions.

Representing detected organisms

To represent the positions of the detected organisms in each video frame, the centroid locations of segmented foreground regions in Cartesian coordinates are used and stored in a vertical cell array matrix, as shown by the parallelogram series (indicating video frames) and Points{t} matrix in Fig. 2. In the Points{t} cell array, the first column stores the temporary identity, numbered from 1 to the number of detected organisms in each video frame, where nobj(t) indicates the number of detected organisms in frame t. The second and third columns store the horizontal and vertical positions of each detected organism in x and y coordinates, respectively. This cell array matrix representation allows for varying element length (indicating the number of detected organisms in frame t), which can change frame-to-frame due to detection and segmentation errors.

Figure 2 Storage structure of detected objects in a video sequence.

The light blue parallelogram series represent the detected organisms within each time-lapse frame. And the cell array Points{t} on the right side shows the detailed data storage structure (that is how the individual identities and object Euclidean locations are arranged) of the detected objects in the example frame t.

Organism assignment between frames

The centroid locations of detected organisms are obtained in the segmentation module and represented frame-by-frame using a cell array for each video sequence as described in the previous section. However, the organism identities over video frames are still unknown. That is, which organism in the current frame corresponds to which organism in the following frame has not been mapped. In addition, there are still some remaining mis-detected organisms that have been classified into background regions or overlapped with other detected organisms. The data association module of MSBOTS thus builds tracking trajectories of individual organisms by mapping the detected organisms between successive pairs of frames, calculating the positions of mis-detected or occluded organisms, and bridging the theoretical locations back to their correct trajectories by adjusting the initial assignment results.

Initial assignment

The initial frame-by-frame data association of detected organisms is a partial assignment using an extension of the Kuhn–Munkres algorithm (Bourgeois & Lassalle, 1971) to process the rectangular cell array Points{t}. Extending on the original Hungarian algorithm (Harold, 1955) that solves the assignment problem with an equal number of workers and tasks represented by a n × n matrix, the number of workers and tasks can be unequal and represented in a rectangular matrix (Bourgeois & Lassalle, 1971). This extended algorithm can then be applied to multiple organism tracking, where the number of detected organisms can change due to non-ideal segmentation resulting from organism mis-detection and occlusion.

In the initial assignment process, all the detected organisms annotated by Points{t} in frame t are taken as source points, and the segmented organisms Points{t + 1} in the following frame t + 1 are seen as the target points. The target points Points{t + 1} need to be mapped to the source points Points{t} frame-by-frame across a video sequence. The Euclidean distance of a source point Si to a target point Tj calculated by Eq. (1) is the cost to connect this point pair. A matrix D, as shown in Eq. (2), is created to represent the cost of assigning source organisms S = {S1, S2, S3,…,Sn } in the frame t to the target organisms T = {T1, T2, T3,, …,Tm } in the frame t + 1.

(1) dSi,Tj=(xj−xi)2+(yj−yi)2

(2) D(S,T)=(dS1,T1dS1,T2dS1,T3⋯dS1,TmdS2,T1dS2,T2dS2,T3⋯dS2,Tm⋮⋮⋱⋮dSn,T1dSn,T2dSn,T3⋯dSn,Tm)

where n and m are the detected number of organisms in the frame t and the successive frame t + 1, respectively.

The frame-to-frame objects mapping searches for unique assignments in the cost matrix D(S,T) by connecting the source organism Si in the frame t to only one target organism Tj in the successive frame t + 1. The sum of the resultant complete assignments between Points{t + 1} and Points{t} is the global optimum with the lowest overall cost amongst all the possible assignments within two successive frames. Thus, the assignment seeks for the most likely correspondences for all detected source points in the successive frame (combinatorial optimisation). The matched target points propagate the identities of the source points. Thus, connecting the points with the same identities across video frames gives the individual organism tracking trajectories after calculating the assignment maps for all the successive video frames.

It is possible to avoid the propagation of false positive detections within the segmentation results when constructing the individual trajectories. In the initial frame-to-frame assignment step, a distance threshold is set as a constraint in the source-target cost matrix D(S,T). The threshold value is calculated by δ * median(dSi,Tj), as in Zhiping & Cheng (2017). When the minimum value of the i-th row in the source-target cost matrix D(S,T) is larger than the estimated threshold value, this indicates that the distances between the source point Si to all of the points in the successive frame exceed the threshold value. Thus, the source point Si is considered as a segmentation noise fragment to not be further assigned to a target point and its corresponding position information will be removed from the point matrix Points{t}. This enables the estimation of reliable tracking results, resilient to ‘bursty’ movement (tested by zebrafish larvae videos as seen in the “Tracking accuracy evaluation” section), flexible and jerky hopping movement patterns (tested by Daphnia videos as seen in the “Tracking accuracy evaluation” section). Only in one occasion where an organism suddenly moves extremely fast in the same direction (resulting in a large distance change within a short period of time over the distance threshold), this process will incorrectly eliminate the organism as a noisy fragment.

When an organism disappears in frame t due to mis-detection or occlusion, a source point in frame t − 1 therefore cannot be assigned to a target. A gap will then occur in the tracking trajectory where the organism fails to be detected, and a new tracking trajectory will start from the frame when the organism is correctly detected again. This source point in frame t − 1 without a mapped target is saved in an unmatched source matrix for this video sequence.

Similarly, when an organism is re-detected in the frame t + n after being missed for n frames, there is one more point in Points{t + n} compared with its previous fame Points{t + n − 1}. To map the points Points{t + n − 1} to Points{t + n}, a point in the frame t + n cannot find a source point in the previous frame. This leftover point in frame t + n is saved in the unmatched target matrix for the same video sequence.

The methods to calculate the theoretical positions between the unmatched source points and unmatched target points, and adding these points to their correct tracking trajectories are explained in the following two sections, respectively.

Position estimation for mis-detected and occluded organisms

Figure 3 illustrates the location computation for the overlapped organism Pi in the frame t + 1. The two points Pi (shown by blue dots) and Pj (shown by green dots) overlap with each other in the frame t + 1, and this overlapped point at time t + 1 is assigned to the object Pj in the initial mapping process. Since the point Pi in the frame t cannot find a target in the frame t + 1, and this point in the frame t + 2 cannot be assigned to a source point in frame t + 1, this point is classified as an unmatched source point in frame t, and an unmatched target point in frame t + 2, respectively. The position of the missing point due to occlusion (as shown by the red dot occluded by the green dot in Fig. 3) or mis-detection (for example, when the red dot is completely occluded by the green dot in Fig. 3) is calculated using the locations of the unmatched source point Pi(t) and unmatched target point Pi(t + 2). In this example, as shown in Fig. 3, the location of the missing point Pi in frame t + 1 is calculated by the median point between Pi(t) and Pi(t + 2).

Figure 3 A point calculation example for occluded organisms.

The object Pi in the frame t +1 as shown by the partial red dot disappears due to occlusion by the object Pj. However, an approximate location of it in the frame t +1 annotated by the orange point can be estimated by the relative locations of its correspondences in the neighbour frames t and t +2.

When there are multiple unmatched pairs, the mapping from unmatched source points to the unmatched target points is also according to the extended Hungarian assignment algorithm (Bourgeois & Lassalle, 1971). In this step, the unmatched source points firstly search for possible correspondences in the following 2nd frame. If no assignment can be mapped, the search extends to the unmatched targets in the following 3rd frame. It was shown in Noss, Lorke & Niehaus (2013) that a trajectory fragment can be bridged to its subsequent tracking fragment so long as the frame separation is less than 6 video frames. Thus, the default search range in MSBOTS is from the 2nd to the 6th frame following the frame when the mis-detection and occlusions originally occur. The positions of missed organisms (xmiss, ymiss) are calculated by Eqs. (4) and (5) using the matched point pair from the the locations of unmatched source point (xs, ys) and an unmatched target point (xt, yt).

(4) xmiss=xs+1j∗(xt−xs)

(5) ymiss=ys+1j∗(yt−ys)

where j indicates the following j-th frame from the unmatched source point.

Bridging trajectory gaps for mis-detected and occluded organisms

Individual tracking trajectories for each organism are obtained through connecting the matched points with the same identities frame-by-frame over a video sequence. However, the tracking trajectories obtained from the initial assignment process are usually trajectory fragments, separated when organisms are mis-detected or overlapped, caused by segmentation errors. In the proposed MSBOTS, these trajectory gaps are bridged by adding the estimated points of these mis-detected or overlapped organisms as described in the previous section.

To connect trajectory fragments, the points stored in the unmatched source matrix are mapped to the points in the unmatched target matrix, and the positions of the missed points between the newly matched unmatched-source to unmatched-target pairs are also calculated during this mapping process. For example, as shown in Fig. 3, the unmatched source Pi(t) as the end point of its trajectory fragment is connected to the unmatched target point Pi(t + 2), which is the start point of its trajectory fragment, and the middle point shown by the orange dot is added between points Pi(t) and Pi(t + 2) as the theoretical position of the overlapped point Pi(t + 1).

Locomotion characteristic analysis

After obtaining the individual tracking trajectories for each organism in the video sequence, the movement characteristics of these organisms can be analysed. The calculation of three movement parameters, movement velocity, acceleration and direction as represented by Eqs. (6–8), respectively, are implemented and presented in this work.

(6) velocity=(xt+1(i)−xt(i))2+(yt+1(i)−yt(i))2dt

(7) acceleration=d(velocity)d2t

(8) direction=atan2yt+1(i)−yt(i)xt+1(i)−xt(i)

where dt = 1/fs and fs is the video frame rate, and xt(i) and yt(i) are the Cartesian coordinates of organism i (i is the assigned organism identity) in frame t.

Results and discussion

To evaluate the performance of MSBOTS, microscopic videos of three types of small biological organisms (zebrafish, Artemia and Daphnia) were tested, where preliminary evaluation using zebrafish was presented in our previous work (Wang et al., 2018; Wang, 2018). Evaluated on a set of single and multiple larvae and adult zebrafish, Artemia and Daphnia videos here, a wide variety of (complex) imaging conditions were tested, including shadowing, labels (manually drawn on the petri dish), and background artefacts (such as water impurities, object faeces and water bubbles of varying sizes). No chemical stimuli were tested on the studied organisms in this work, so their behaviour analysed corresponds to natural response. In addition to the tracking accuracy evaluation, the natural locomotive characteristics as described by movement velocity, acceleration and direction are also analysed on the video datasets to test the dynamic behaviour analysis capability of the proposed system.

Small biological organism datasets

Microscopic time-lapse videos of three types of small organism models: zebrafish, Artemia and Daphnia were applied to evaluate the proposed MSBOTS platform. Both low frame rate videos (of 14 or 15 fps recorded by a AD7013MT Dino-Lite microscope) and high frame rate videos (captured by an UI-3360CP-C-HQ microscope) are tested in this work.

Wild zebrafish (Danio rerio) embryos were incubated at 28 °C in a petri dish filled with an E3 medium. Zebrafish larvae were obtained from the hatched embryos 5 days post-fertilization. Zebrafish larvae were transferred to round poly (methyl methacrylate) (PMMA) housing wells for shooting microscopic time-lapse videos. The zebrafish dataset consists of 10 video sequences with 3,056 frames in total: nine zebrafish larvae videos captured as prescribed here and one adult zebrafish video provided by Pérez-Escudero et al. (2014) from their publicly available online repository (‘Example video of 5 zebrafish’ at http://www.idtracker.es/download). The size of the zebrafish larvae video frames is 960 pixels × 1280 pixels, and the average zebrafish larvae size is around 1900 pixels. More details about the larvae zebrafish dataset and its generation can be found in our previous work (Wang et al., 2017b).

Cysts of the marine crustacean Artemia franciscana and freshwater Daphnia magna were hatched and cultured according to the Artoxkit-M and Daphtoxkit-F (MicroBioTests Inc., Gent, Belgium) standard operating protocols. Artemia franciscana were hatched in a petri dish filled with sea water (pH 8.0 ± 0.5) at 24 ± 0.5 °C under exposure to 3,000–4,000 lux light source for 30 h. Artemia were placed into a group of 10 in a miniaturised Lab-on-a-Chip (LOC) chamber as per Solis et al. (2015) when shooting videos with microfluidics infused at a flow rate of 5.25 mL/h. Five Daphnia magna neonates were randomly selected and transferred into a petri dish with the temperature maintained at 20.0 ± 0.5 °C. The Artemia franciscana and Daphnia magna dataset consists of 5 video sequences each with 4,802 frames and 4,804 frames in total, respectively.

Artemia franciscana microscopic videos containing five organisms and artifacts (bubbles of different sizes, video sequences 1–5), with Fig. 4B a microscopic video frame example of Artemia franciscana in 480 pixels × 640 pixels. The average size of an Artemia franciscana object is approximately 500 pixels. Daphnia magna microscopic videos containing 10 organisms and artifacts (bubbles and impurities of different sizes, video sequences 1–5), with Fig. 4C a microscopic video frame example of Daphnia magna in 480 pixels × 640 pixels. The average size of Daphnia magna tested is approximately 400 pixels. Both the Artemia and Daphnia videos were ordered randomly to test the flexibility and feasibility of the proposed system.

Figure 4 Microscopic video frame examples of (A) zebrafish larvae (960 pixels × 1280 pixels), (B) Artemia franciscana (480 pixels × 640 pixels), and (C) Daphnia magna (480 pixels × 640 pixels).

The average sizes of zebrafish larvae, Artemia and Daphnia tested are approximately 1,900, 500 and 400 pixels, respectively.

Tracking evaluation metrics

To objectively and quantitatively evaluate the tracking performance of the proposed MSBOTS platform, the widely utilised standard metric, Classification of Events, Activities and Relationships (CLEAR MOT) (Bernardin & Stiefelhagen, 2008), for Multiple Object Tracking (MOT) is employed in this paper. CLEAR MOT comprises two metrics: Multiple Object Tracking Precision (MOTP) as presented by Eq. (9) and Multiple Object Tracking Accuracy (MOTA) as shown by Eq. (10). MOTP measures the position precision of all segmented organisms compared to that of the manually labelled ground truth in every video frame, whereas MOTA estimates the individual trajectory accuracy (the ability to produce exactly one trajectory per organism with a consistent label over time).

(9) MOTP=∑i,t|Pi,t−GTi,t|∑tNt

(10) MOTA=1−∑t(Mt+FPt+SIIt)∑tgt

where |Pi,t − GTi,t| is the Euclidian distance between the estimated centorid position of the i-th detected organism Pi(t) in the frame t. Its position in the manually labeled ground truth is denoated as (GTi,t) and Nt indicates the total number of segmented organisms in the ground truth in frame t. The smaller the MOTP value, the more precise the segmentation result.

MOTP measures the organism segmentation accuracy compared with the ground truth, whereas the MOTA metric emphasises the evaluation of the individual tracking trajectories, where Mt, FPt, and SIIt indicate the number of missed detections, false positive detection (i.e., image noise fragments segmented as organisms), and the Swapping of Individual Identities (SII), respectively, in the frame t. And gt means the total number of organisms detected in frame t. The ideal value of MOTA is 1, and the value decreases with the occurrence of detection errors and identity swapping. Comparing with the total number of detected organisms, the MOTA value will generate negative numbers (as shown in Fig. 5A) when the combination of detection errors and identity swapping is high, which implies low system reliability and the resultant tracking trajectories should be considered as unreliable.

Figure 5 Evaluation of tracking results comparing the proposed system MSBOTS to existing systems - SimpleTracker (Bourgeois & Lassalle, 1971), idTracker (Pérez-Escudero et al., 2014) and LoliTrack (Závorka et al., 2017)—using videos of three types of small biological organism.

(A–C) The estimated MOTP and MOTA values for zebrafish, Artemia and Daphnia video datasets, respectively. MOTP values measure the segmentation performance, and the smaller the value the higher the segmentation accuracy. MOTA compares the accuracy of the resultant individual tracking trajectories,with a higher value denoting a more accurate result.

Tracking accuracy evaluation

To evaluate the proposed tracking system, its tracking performance over the microscopic video dataset is compared with the well-known multiple object tracking platform idTracker (Pérez-Escudero et al., 2014), SimpleTracker (Bourgeois & Lassalle, 1971) (using the Kuhn-Munkres tracking algorithm for initial association and the nearest neighbourhood to directly connect trajectory fragments, without considering missed organisms), and the off-the-shelf commercial LoliTrack system (Závorka et al., 2017). Fig. 5 shows the estimated tracking accuracy measured by the MOTP and MOTA metrics, with Fig. 5A showing results from zebrafish videos, Fig. 5B showing Artemia videos, and Fig. 5C showing Daphnia videos.

It can be seen from Fig. 5A (the upper MOTP figure) that the positions of detected organisms using adult and larvae zebrafish videos consistently exhibit the smallest distance differences with the manually labelled ground truth positions amongst all the methods tested. The proposed MSBOTS approach resulted in overall smaller MOTP values of 0.92, 25.59, and 44.48 pixels (71.59% increase) compared to SimpleTracker, idTracker, and LoliTrack, respectively. The improved organism position detection results demonstrate the accuracy of the theoretical position estimation based on the organism detection and segmentation results using the adaptive GMM model and post-processing in the proposed MSBOTS system. Table S1 summarises the estimated tracking accuracy measured by MOTP and MOTA values across all the zebrafish video sequences.

All of the methods performed well for the position accuracy of detected organisms when the videos had a clear background, as shown by sequences 1–6. However, the location errors measured by MOTP for the detected organisms in the zebrafish videos compared to the ground-truth did not decrease as dramatically as LoliTrack or idTracker with increasingly complex video backgrounds, as shown by the MOTP values in sequences 7–10 in Fig. 5A. Accordingly, the proposed MSBOTS method still out-performed the existing approaches when taking into account mis-detection, false positive segmentation and identity swapping, with 31.20%, 63.01%, and 24.61% higher MOTA values than SimpleTracker, idTracker, and LoliTrack, respectively. This was mainly achieved by the ability to estimate the positions of the mis-detected or overlapped organisms using their neighbour position knowledge from the segmentation results in the proposed MSBOTS platform. In addition, the bridging of trajectory fragments in MSBOTS based on the extended Hungarian assignment algorithm (Bourgeois & Lassalle, 1971) when there are multiple unmatched trajectory fragments decreased the possibility of individual identity swapping, compared with SimpleTracker, which only used a distance metric by nearest neighbour algorithm (which in turn can generate identity swapping during the gap bridging process).

In addition, Figs. 5B and 5C show the tracking accuracy evaluation using Artemia franciscana and freshwater Daphnia magna videos, respectively, to test the generalisation of the proposed MSBOTS system on other small organisms with movement characteristics different to zebrafish larvae. Artemia display flexible movement and vary according to surrounding fluidics (Tyson, 1974; Williams, 1994), and Daphnia exhibits short, jerky hopping movement in water (Rottmann et al., 1992). The overall tracking accuracy performance of the proposed MSBOTS method is consistent for the tested videos on these two organism types. The detailed data on tracking accuracy measured by MOTP and MOTA values can be seen in Tables S2 and S3.

As can be seen from the MOTP values in Fig. 5B, the proposed method exhibits 47.68 pixels smaller standard deviation than the idTracker system, which illustrated the usability of the proposed system on Artemia microscopic videos and the enhanced ability on Artemia detection accuracy. Though idTracker can produce smaller organism position estimation errors (MOTP) as shown by sequences 3–5 in Fig. 5B, the mean MOTA value is 7.07%, and 6.44% less than the proposed MSBOTS and LoliTrack, respectively, which illustrates the existence of a similar detection problem when testing zebrafish larvae. That is, the organism is detected as background, and impurities as an organism, due to their small size differentiation and similar movement characteristics when the organism stops moving or the water impurities are stirred up by organism movement; this further causes identity confusion.

Figure 5C shows the overall tracking accuracy of the proposed MSBOTS system and idTracker consistently outperformed the other two systems under comparison, measured by MOTP and MOTA. However, the mean MOTA value of the proposed MSBOTS system is still 5.48% higher than idTracker. The generalised tracking accuracy of the proposed MSBOTS system is further illustrated by applying it to Daphnia video sequences with short, jerky hopping movement characteristics, compared with these existing systems.

Analysis of organism movement characteristics

To explore the capability of the proposed MSBOTS approach in analysing organism movement characteristics, this paper presents the estimation of movement velocity, acceleration and direction calculated from the individual tracking trajectories.

Figure 6 shows the movement acceleration analysis for each zebrafish video using the resultant tracking trajectories generated by the proposed MSBOTS. It was found that in Hinz & de Polavieja (2017), the interaction and movement of zebrafish larvae was very close to zero by 7 dpf. As shown in Fig. 6, in general the variation of movement speed (shown by the computed acceleration values) that is obviously visually perceptible to the human eye occurred only within approximately 10 s, as tested in the 10 adult and larvae zebrafish videos. This is due to the natural (anxious) response of zebrafish when turning on the imaging camera (Peng et al., 2016). The zebrafish movement speeds are more subtle from this point on, which is consistent with the zebrafish movement characteristics found in Peng et al. (2016); Hinz & de Polavieja (2017).

Figure 6 Organism movement acceleration over time for every tested zebrafish video (one video per subplot), with the centre subplot combining the results of videos 5 and 6 housing a single zebrafish larvae each.

The line colors represent different zebrafish individuals in every video.

As organism movement speed and direction changes can provide insight into the interaction rules (Hinz & de Polavieja, 2017), Fig. 7 shows an example of the calculated velocity and movement direction results for each zebrafish video, respectively. The equal number of boxes with the number of zebrafish housing in each video show that all individuals were successfully assigned with one identity (which also illustrated the one-to-one organism mapping accuracy in the proposed MSBOTS system), and the median velocity (labelled by the red line inside each box) of each larvae indicates the consistent movement characteristic within the same housing well. Mean, minimum and maximum velocity values can also be easily obtained from the visual box plot shown in Fig. 7.

Figure 7 Analysis of zebrafish locomotion characteristics across video sequences, where (A) and (B) show movement speed and direction analysis of individual organisms, respectively, in very video frame (the centre subplots combines the results of single zebrafish larvae in videos 5 and 6).

There is no specific speed requirement for the organisms that can be tracked by the MSBOTS platform. Both organisms travelling at high speed or average low speed as illustrated by Figs. 6 and 7A, respectively, can be accurately detected and tracked over time by the system. Thus, this system can be used for automatic tracking, comparison and analysis of small organisms in natural response or under the exposure of testing chemicals. The recommended lowest frame rate of time-lapse videos is 14 f/s as in the tested dataset from trial-and-error comparison.

Conclusion

Accurate automatic tracking for multiple small biological organisms provides an efficient approach for many biomedical and ecotoxicity applications. However, organism mis-detection and occlusion are inevitable problems when detecting and segmenting these small biological organisms from time-lapse microscopic videos. The detection and segmentation becomes more challenging when tracking small biological aquatic organisms compared with general objects, which in turn affects the subsequent organism tracking processes. To improve the tracking accuracy based on the non-ideal organism detection and segmentation results, extending on and improving our previous work, this paper presents a Multiple Small Biological Organism Tracking System (MSBOTS), combining a multiple object association algorithm for linking detected objects frame-by-frame and tracking trajectory adjustment techniques. To address segmentation errors due to mis-detected or occluded organisms, the proposed MSBOTS approach estimated positions of organisms in interim frames using corresponding points in neighbouring frames. Finally, the calculated points are applied to connect and adjust the tracking trajectory fragments from the initial association based on an extended Kuhn–Munkres algorithm. The proposed system was tested on three different types of small organisms with variant movement characteristics, using 20 videos in total for evaluation. The resulted tracking accuracy of the proposed system outperformed three existing (state-of-the-art or commercial) tracking systems. Moreover, this system also provides locomotive characteristic analysis using the generated individual tracking trajectories to facilitate small organism behaviour analysis research. Behavioural rules and new medicine or chemical effects on the dynamic behaviour of organisms can thus be investigated using the proposed behaviour analysis module, enabling an automatic and quantitative movement analysis to the related experiments.

Supplemental Information

Supplemental Information 1 Tracking performance comparison among the evaluated systems testing on zebrafish time-lapse video dataset.

Click here for additional data file.

Supplemental Information 2 Tracking performance comparison among the evaluated systems testing on artemia time-lapse video dataset.

Click here for additional data file.

Supplemental Information 3 Tracking performance comparison among the evaluated systems testing on daphnia time-lapse video dataset.

Click here for additional data file.

I acknowledge my current supervisor’s support (Prof. Mark Eston) in permitting me to continue working on this article and the research behind it.

Additional Information and Declarations

Competing Interests

Author Contributions

Data Availability

The authors declare that they have no competing interests.

Xiaoying Wang conceived and designed the experiments, performed the experiments, analyzed the data, prepared figures and/or tables, authored or reviewed drafts of the paper, and approved the final draft.

Eva Cheng conceived and designed the experiments, authored or reviewed drafts of the paper, and approved the final draft.

Ian S. Burnett conceived and designed the experiments, authored or reviewed drafts of the paper, and approved the final draft.

The following information was supplied regarding data availability:

The datasets, developed MSBOTS system, and the software to analyse locomotive characteristics of individual organisms are available at GitHub: https://github.com/Xiao-ying/moving-zebrafish-larvae-segmentation-and-tracking-dataset-/tree/master/Data.

The analysed adult zebrafish video (‘Example video of 5 zebrafish’) is available at idTracker.

http://www.idtracker.es/download.

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
