# Peer review of "MSBOTS: a multiple small biological organism tracking system robust against non-ideal detection and segmentation conditions"

_PeerJ, doi:10.7717/peerj.11750_

## Round 0.1 · original submission · Major Revisions

Both reviewers have provided constructive and actionable comments. Please address each comment specifically, indicating how it was addressed in the revised manuscript. I encourage you to pay particular attention to questions about the videos provided on the Github repository and provide further information regarding the ethical requirements for working with adult zebrafish.

Reviewer 1 ·

Basic reporting

Figure captions should be improved and axis-labels clearly explained.

Experimental design

Methods description needs some improvement.

Validity of the findings

No comment

Additional comments

The authors present a fairly detailed study of the development of a tracking system for multiple small biological organism. There are some gaps in the study and/or presentation of the research. The comments below are separated as major or minor.

Major Comments:
1) Line 139: what is a decaying envelope factor (alpha)? Also, it is not clear whether the method is suitable for detection and tracking if the organisms can stop at a location for a considerable amount of time before restarting their motion.
2) Line 197-198: what is the underlying idea or reason for optimizing assignment based on the lowest summed distance within two successive frames?
3) Line 202-209: to eliminate the false positives when constructing an individual trajectory, a method is described which removes points as segmentation noise based on a threshold distance criterion. Would this approach produce good results if the velocities are spread over a wide range (say, an order of magnitude), or would this method produce reliable results only when organisms are moving at similar values of velocities.
4) Is the method robust in situations where an organism can move both forward and backward?
5) Figure 4: The caption should be more descriptive (even though description is provided in the text when discussing this figure). What are (a), (b), and (c)? What is sequence? If sequence cannot have non-integer values, then why show them? Is the Simple Tracker the manual connection of location?
6) The measure of MOTP is the number of pixels. This may not help readers fully understand the tracking performance if two important information are not known: (a) average number of pixels an organism occupies? and (b) the size of the video frame in pixels?
7) The MOTA measure should be discussed in more detail to help the readers develop a better intuitive understand. Based on Eq. (10), the ideal value of MOTA is 1. What does a negative value of MOTA (as shown in Fig. 4a) imply?
8) Fig. 6 and 7: Moe description in the figure caption is needed. What’s the difference between the 9 boxes in Fig. 6? What are the colors representing in Fig. 6? Why is the unit of acceleration shown as pixel and not pixel^2?

Minor Comments:
1) MOTP and MOTA are used in the abstract without defining it first.
2) For computation of organisms’ centroid, details should be provided about the recommended video resolution (i.e., minimum number of pixels for each organism), and how the centroid computation is handled when two or more organisms are in very close proximity.
3) Details need to be added on the average speed of organisms being tracked and the recommendations for corresponding minimum frame rate.

·

Basic reporting

The writing is not always clear, with a number of areas in the manuscript where the English grammar needs to be improved. I have attached an annotated pdf where I have highlighted the issues.

The figures should stand alone, and they do not - the figures legends barely provide any information. The reader is forced to read the text to make sense of the figures. Suggestions for improvement can be found in general comments below.

Experimental design

The authors compare their tracking software to three others, and I would think that this would involve some statistical comparison, but the authors only seem to report the % difference, which I presume is across all the video sequences? It's certainly possibly to report standard error or confidence intervals around this mean, but the video sequences are not technically replicates, as they increase in complexity. Perhaps this is OK for this type of study though.

I am not convinced that the study is repeatable, because there is not enough information on the complexity of each video (see further comments below).

I am also not sure that this manuscript meets ethical standards - the authors state that ethics were not required, but they used adult zebrafish, so I assume that would have needed ethical approval for this. Perhaps this just requires a bit more clarification. In general, it's not entirely clear if adult videos were used - they are mentioned several times int the manuscript, but I don't see any mention of adults on the github repository.

Validity of the findings

I was not able to find the zebrafish image frames on the github site, even though I was in the correct project folder.

Additional comments

This manuscript proposes a new organism tracking system, MSBOTS, and uses a series of videos to compare the performance of MSBOTS and three other tracking systems. Most of the methods contain a lot of technical jargon, so I did find it hard to follow, but I could understand the results and figures (once I deciphered what each figure was showing, as the figure legends provide almost no detail). MSBOTS appears to perform well, and better than other systems when the video frames have things like bubbles, shadows, debris, etc (i.e., later video sequences). That said, there are many aspects of the manuscript that can be improved, particularly the presentation of the figures. I also think there needs to be a clear justification on what this paper adds in terms of the zebrafish aspect of the study. Is this the same analysis as presented in the previous published study, or is this new? I also question the statement that ethical approval was not applicable because no chemicals or medicines were being tested. No ethics are required for the invertebrates that are used, and not the zebrafish larvae, but any manipulation of adults (even netting into a different tank) would require ethical approval in Australia.

Comments to help improve figures:
Figure 4. Why not label the blue ‘proposed’ as MSBOTS? The legend for this figure needs to define the axis. Otherwise, the reader needs to try and work out what MOTP and MOTA stand for – figures should stand alone. The symbol legend for panel b seems to have moved as well. It’s also a bit odd to put the a, b and c labels below the figures and the panels are not defined anywhere, except in the text. It is not clear why the x-axis includes 1.5, 2.5, 3.5, etc. for panels b and c. This represents the 5 and 5 videos sequences, so should be labelled with whole numbers, as in panel a. Also, there should be no connecting lines between the points – the sequences are different videos (if I understand the design correctly), so should be represented as point values only. While I recognize that the lines help to illustrate the patterns, it is technically misleading to include them.

Figure 4 appears well before it is ever mentioned in the text, and figure 5 appears before figure 4.

Figure 5 panel b and c are quite blurry, the artemia and rotifer within anyways. This figure would be more meaningful if it showed a frame from the least complex and most complex video for each species.

No idea what all the panels in Figure 6 are. I think I finally inferred that they are the zebrafish videos, 1-10, with 5 and 6 combined, as in Figure 7. But what are all the colours, and what scale is time on, seconds?

Figure 7, why are videos 5 and 6 labelled with video 5 and 6 on the x-axis? To clarify, video 1 had 4 objects, video 2 had 3, video 4, had 5? Some of the videos have very high velocities compared to others? What are all the red crosses in the figures? Are these outliers? If so, these should be a different colour or symbol (open circles work well for outliers), as it detracts from the red line used to illustrate the median.

For Figures 6 and 7, simplify the figures – remove the y and x labels from all except the far left and bottom panels and remove the O from the object labels on the x-axis for Figure 7 – simply label as 1, 2, 3, etc...

Figure 6 and 7 are zebrafish only? This should be indicated in the figure legend – again, figure legends need to stand alone. Not entirely clear what figures 6 and 7 are adding to the study, in terms of displaying the data for each figure separately? Would it be better to summarise the data in some way, or perhaps these be better suited to the supplement?

Other comments:
The description of the data on the github repository is a bit unclear. It says “this dataset consists of 10 video sequences....(including zebrafish larvae, Artemia and Daphnia)”, but it’s actually 5 artemia videos and 5 daphnia videos in this MSBOTS project folder, so the authors just need to remove zebrafish larvae from this first statement. They then detail where the zebrafish larvae (should this be adults too, as adults are mentioned several times in the manuscript text?) can be found, in the moving zebrafish larvae segmentation and tracking dataset. However, I can’t seem to access the zipped video files for the zebrafish though.

As for the artemia videos, video 1 appears to be the more complex (includes bubbles) in the sequences for artemia, but I infer that 5 should be most complex from the manuscript text, as that is how it is set up for the zebrafish? Not clear how the daphnia videos differ – video frames from video 1 and 5 look the same? Again, as mentioned above, more detail on the video settings is needed, in order to make this study repeatable.

Line 78. I have used idTracker in the past, and we didn’t find that the image conditions were strictly constrained any more than any other tracking program, so perhaps clarify this statement a bit further.

Not clear why there are so many methods in the results and discussion section, lines 268-312?

Line 279: drasophila? I think you mean artemia!!

If adult zebrafish are included in this study, then there needs to be more detail (i.e., fish age, sex, line, tank dimensions).

Definitions for MOTP and MOTA do not appear till the section that is at line 306.

Line 350 refers to supplementary table S2 and S3, but these are not included for review. There also is no reference to table S1?

The references are inserted incorrectly, without commas or brackets to separate them from the text, which made it quite tricky to read the manuscript.

Minor comments:
Figure legend 1 should include ‘biological’

Line 126. What is the unit for the 0.075 value

Line 131. ‘the’ should be removed

Line 133. Comma required after ‘approach’

Lines 302 to 305, not sure why these are presented as bullet point – incorporate into text. Note that the word examples in these should be example.

Line 410, depository should be repository

---

## Round 0.2 · accepted · Accept

Thank you for your responses and careful attention to addressing all reviewer comments. I have reviewed the responses and the modified manuscript and in my opinion, the review comments have been fully and adequately addressed. I recommend that the paper be accepted for publication.